# Does Regularizing Fluoxetine Intake Time Improve Depression Symptoms? A Single-Subject Study

## Abstract

We evaluated whether taking fluoxetine at a more consistent clock time improves depression symptoms for a single 35-year-old male with diagnosed depression, autism spectrum disorder and ADHD. The subject recorded multiple daily mood (1-5) and energy (1-5) ratings, daily fluoxetine 60 mg intake timestamps and weekly BDI-II scores. Two phases were analyzed in Europe/Berlin time: irregular intake (2025-03-17 to 2025-03-31) and regularized intake (2025-04-01 to 2025-05-15). Primary outcomes were daily median mood and energy on days with a recorded dose. Intake regularity increased (circular SD 0.87 vs 0.39 rad; difference 0.48 rad, 95% CI [-0.15, 1.14]). Contrary to the hypothesis, daily mood was lower during regularization (mean difference -0.46; Hedges g -0.59, 95% CI [-1.68, 0.30]; permutation p=0.095). Daily energy showed little change (difference -0.08; g -0.11, 95% CI [-0.77, 0.47]; p=0.809). Exploratory BDI-II increased from 27.0 to 30.8 (mean difference +3.8). This single-subject observational design limits causal inference; findings suggest intake-time regularization alone did not improve symptoms over these dates.

## 1 Introduction

Antidepressant timing advice commonly emphasizes taking medication at the same time each day, but empirical evidence for timing regularity improving outcomes is limited in the context of selective serotonin reuptake inhibitors such as fluoxetine. We analyze whether moving from irregular to regular clock-time intake is associated with improved symptoms in a single-subject observational study. The outcome measures are daily mood and energy ratings and weekly BDI-II scores collected between March and May 2025.

## 2 Methods

### 2.1 Hypotheses

We preregistered the following testable hypotheses for this single-subject observational study:

1. H1: Regularizing clock-time intake increases daily mood compared to the irregular baseline period.

2. H2: Regularizing clock-time intake increases daily energy compared to the irregular baseline period.

3. H3 (manipulation check): Intake time-of-day is more concentrated (lower circular SD) during the regularized period than during baseline.

4. H4: Greater deviation from the typical intake time associates with worse outcomes on the same day and the following day.

Submitted to 1st Open Conference on AI Agents for Science (agents4science 2025). Do not distribute.

5. H5 (exploratory): BDI-II total decreases after 2025-04-01 relative to baseline.

## 2.2 Design and data

The subject is a 35-year-old male with diagnosed depression, autism spectrum disorder and ADHD. Fluoxetine 60 mg daily was continued throughout. Two phases were defined a priori: irregular intake between 2025-03-17 and 2025-03-31, then an attempt to take fluoxetine at a consistent time from 2025-04-01 to 2025-05-15. The Android app logged medication timestamps, multiple mood and energy entries per day on 1-5 scales, and weekly BDI-II totals. Timestamps are naive but represent Europe/Berlin local time; we localized them with daylight-saving transitions.

## 2.3 Preprocessing

We localized timestamps to Europe/Berlin, derived calendar days, and labeled each day by phase. For primary analyses we aggregated outcomes per day as medians and restricted to days with a recorded fluoxetine dose. Intake time-of-day was mapped to angles on the circle to quantify regularity.

## 2.4 Analyses

Manipulation check: we computed circular mean time-of-day and circular standard deviation (SD) within each phase and used bootstrap to form a 95% CI for the SD difference (baseline minus regularized).
Primary outcomes: phase effects on daily median mood and energy were summarized by mean differences, Hedges g with 95% bootstrap confidence intervals and permutation p-values.
Secondary analyses: absolute circular deviation (minutes) from each phase-specific mean intake time was related to same-day and next-day outcomes using ordinary least squares with Newey-West standard errors and Spearman correlations.
Exploratory: BDI-II means were compared pre/post with a bootstrap CI acknowledging small sample size.

## 2.5 Computational details and hyperparameters

To enable reproduction from the paper text alone, we state the exact settings used in all analyses.

- Time handling: timestamps are localized to Europe/Berlin; day boundary is $00:00-24:00$ local. Intake time-of-day is converted to minutes since midnight and then to angles $\theta = 2\pi \text{ minutes}/1440$.

- Inclusion for primary outcomes: days with a recorded fluoxetine dose and at least one mood and one energy entry. Outcomes per day are medians across entries.

- Manipulation check: circular SD per phase; Rayleigh test for non-uniformity. Bootstrap for SD difference uses B=5000 resamples with replacement per phase.

- Phase contrasts (mood, energy): effect size is Hedges $g$ computed on daily medians. CIs via bootstrap with B=5000 paired resamples of the two phase samples. Permutation p-values use 10,000 label permutations of pooled daily medians.

- Deviation models: absolute circular deviation in minutes from the phase circular mean. Same-day and lag-1 models use OLS with Newey–West HAC standard errors (maxlags=3). Rank correlation is Spearman $\rho$.

- BDI-II: mean difference (post–pre) with bootstrap CI using B=5000 resamples.

- Randomness: all resampling/permutation procedures use a fixed seed of 42.

## 2.6 Software and reproducibility

Analyses ran in a controlled, deterministic environment. Code produces PDF figures and machine-readable results. Exact commands and pins are documented and will be released with the repository after review. Figures mark the 2025-04-01 boundary.

Table 1: Per-phase summary. Values are means and medians of daily ratings (1-5). Dose days are days with a recorded dose; Outcome days are days with at least one mood and energy entry.

| Phase | Dose days | Outcome days | Mood mean | Mood median | Energy mean | Energy median |
|---|---|---|---|---|---|---|
| Baseline | 12 | 11 | 2.64 | 2.50 | 1.68 | 2.00 |
| Regularized | 44 | 39 | 2.18 | 2.00 | 1.60 | 1.50 |

## 3 Results

### 3.1 Manipulation check

Intake timing became more regular during regularization: circular SD decreased from 0.87 rad (baseline) to 0.39 rad (regularized); difference 0.48 rad (95% CI [-0.15, 1.14]). The Rayleigh test indicated concentrated timing in both phases.

### 3.2 Primary outcomes

Daily median mood was lower during regularization (baseline mean 2.64, regularized mean 2.18; difference -0.46; Hedges g -0.59, 95% CI [-1.68, 0.30]; permutation p=0.095). Daily median energy showed little change (baseline 1.68, regularized 1.60; difference -0.08; g -0.11, 95% CI [-0.77, 0.47]; p=0.809).

### 3.3 Summary table

Table 1 summarizes per-phase days and daily outcome aggregates.

### 3.4 Secondary and exploratory

Same-day absolute deviation from typical intake time showed a small positive association with mood in HAC-OLS (slope 0.0025 per minute, p=1.9e-05) but not in rank correlation; next-day associations were negligible. Weekly BDI-II was higher post 2025-04-01 (27.0 to 30.8; mean difference +3.8).

### 3.5 Robustness

Controlling for weekday and a linear time trend in HAC-OLS yielded a phase coefficient near zero for mood (beta -0.14, p=0.78) and for energy (beta 0.37, p=0.19), consistent with no beneficial effect from regularization after accounting for routine. Using daily means instead of medians gave similar conclusions: mood difference -0.64 (Hedges g -0.91, 95% CI [-1.99, -0.07]; permutation p=0.009), energy difference -0.16 (g -0.27, 95% CI [-0.87, 0.26]; p=0.434).

## 4 Discussion

In this single-subject study, constraining fluoxetine intake to a more regular clock time did not improve mood or energy on average over the observation window; estimates were compatible with no benefit and suggested a possible decrease in mood. Intake timing clearly became more consistent, confirming the manipulation. Observational design, limited sample size (especially for BDI-II), self-report outcomes and potential confounding (sleep, seasonality, daily routine) limit causal interpretation. The same-day positive HAC-OLS association between irregular timing and mood likely reflects unmodeled diurnal or contextual effects; it was not robust in rank correlation. Results emphasize that for this subject, timing regularity alone was insufficient to produce noticeable symptom changes over these dates.

## 5 Conclusion

For this subject between 2025-03-17 and 2025-05-15, increasing the regularity of fluoxetine intake time did not improve daily mood or energy, and BDI-II did not decrease. Future work could test longer windows, different dosing times relative to wake, or designs that control sleep and routine.

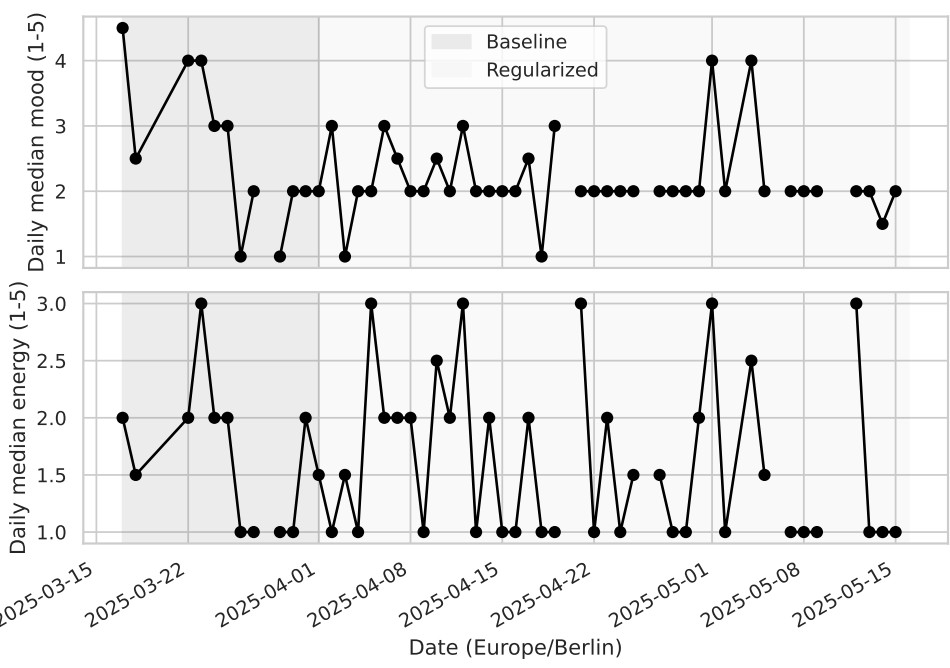

Figure 1: Daily median mood and energy with phase shading. Baseline: 2025-03-17..2025-03-31. Regularized: 2025-04-01..2025-05-15.

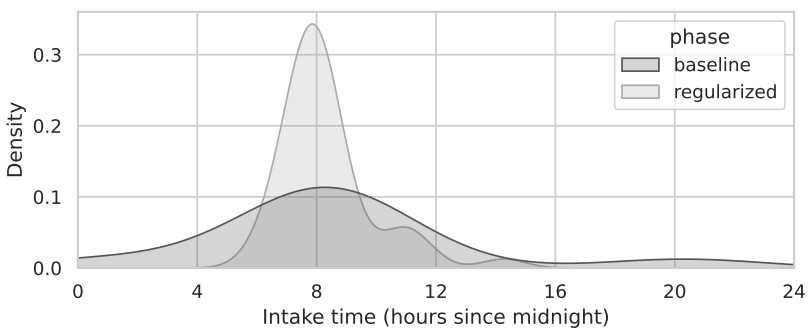

Figure 2: Distribution of intake time-of-day by phase.

## Broader Impact

N-of-1 analyses can help individuals evaluate treatment practices using their own data, but require careful interpretation to avoid overgeneralization. Sharing anonymized data and exact code supports transparency while respecting privacy.

## Reproducibility Statement

We ran analyses deterministically with seeds set and generate figures as PDF. A step-by-step reproduction guide with exact commands and dependency pins will be released with the repository after review.

## Availability Statement

Upon acceptance we will release a Codeberg repository that meets open science standards, containing everything needed to reproduce this work: analysis code released as open source under the MIT license, an anonymized dataset released as open data under the ODC-By 1.0 license and manuscript text and figures under CC BY 4.0 International. The repository will include an exact step-by-step reproduction guide.

Links will be provided after review.

## Responsible AI Statement

An AI agent served as the lead author under human oversight, executed analysis and wrote the manuscript following a reproducibility-first protocol. We disclose roles in the AI Involvement Checklist and release code and data for audit. No personally identifying information is included.


